# EGODEX: LEARNING DEXTEROUS MANIPULATION FROM LARGE-SCALE EGOCENTRIC VIDEO

**Ryan Hoque**[*], **Peide Huang**[*], **David J. Yoon**[*], **Mouli Sivapurapu, Jian Zhang**

Apple

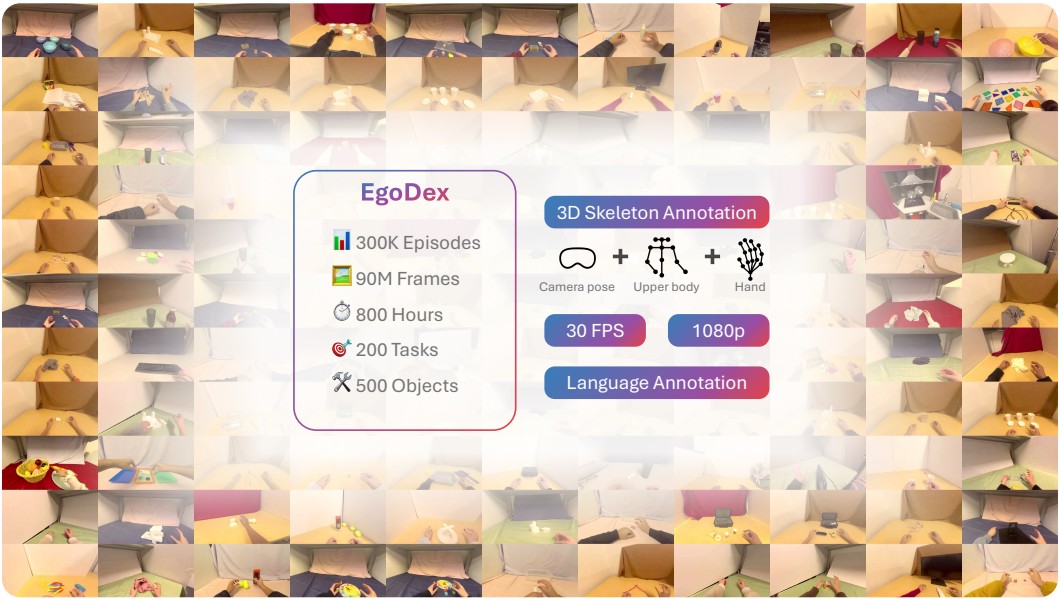

Figure 1: **EgoDex** is a large-scale egocentric dataset that focuses on human dexterous manipulation.

## ABSTRACT

Imitation learning for manipulation has a well-known data scarcity problem. Unlike natural language and 2D computer vision, there is no Internet-scale corpus of data for dexterous manipulation. One appealing option is egocentric human video, a passively scalable data source. However, existing large-scale datasets such as Ego4D do not have native hand pose annotations and do not focus on object manipulation. To this end, we use Apple Vision Pro to collect EgoDex: the largest and most diverse dataset of dexterous human manipulation to date. EgoDex has 829 hours of egocentric video with paired 3D hand and finger tracking data collected at the time of recording, where multiple calibrated cameras and on-device SLAM can be used to precisely track the pose of every joint of each hand. The dataset covers a wide range of diverse manipulation behaviors with everyday household objects in 194 different tabletop tasks ranging from tying shoelaces to folding laundry. Furthermore, we train and systematically evaluate imitation learning policies for hand trajectory prediction on the dataset, introducing metrics and benchmarks for measuring progress in this increasingly important area. By releasing this large-scale dataset, we hope to push the frontier of robotics, computer vision, and foundation models. EgoDex is publicly available for download at `https://github.com/apple/ml-egodex`.

---

[*]Equal contribution.

# 1 INTRODUCTION

The "bitter lesson" (Sutton, 2019) of recent breakthroughs in large language models and large vision models is that the simple recipe of supervised learning with vast amounts of data is far more effective than competing approaches. Two key challenges have prevented the application of the bitter lesson to the longstanding challenge of autonomous robot manipulation: (1) it is unclear what data should be collected, and (2) it is unclear how such data can be collected at the requisite scale.

The leading approach to data collection for robot imitation learning is teleoperation, in which human operators provide demonstrations by directly controlling robot hardware. Recent works such as Open X-Embodiment (Padalkar et al., 2024) and DROID (Khazatsky et al., 2024) pioneer community-wide efforts to pool together hundreds of hours of teleoperation data. While such datasets can be used for pretraining robot control policies, teleoperation is bottlenecked by physical robot operation, and it is unclear how to continue scaling this paradigm beyond its current size. Other works explore learning visual representations from existing in-the-wild Internet videos and images (Radosavovic et al., 2023; Ma et al., 2023). In this case, while large-scale data is available, unstructured video data lacks the precise annotation necessary to learn dexterous manipulation.

We explore a middle path between the two: egocentric human video with paired 3D hand pose annotations. As suggested by recent work (Kareer et al., 2024; Qiu et al., 2025), such an approach is *passively scalable*, similar to text and images on the Internet. Effectively learning from such data is critical in a future where wearable headsets and smart glasses may be omnipresent. Data is a crucial component for doing so; before AlexNet (Krizhevsky et al., 2012) must come ImageNet (Russakovsky et al., 2015).

To this end, we introduce EgoDex: a large-scale dataset and benchmark for learning dexterous manipulation from large-scale egocentric video. EgoDex consists of 829 hours of 30 FPS video and paired skeletal data with a total of 90 million frames and 338000 task demonstrations across 194 tabletop manipulation tasks. To our knowledge, the EgoDex dataset is the largest and most diverse dataset of dexterous human manipulation to date.

There are several key properties of the proposed data that make it more suitable for dexterous manipulation than existing alternatives:

- EgoDex is passively scalable, unlike robot teleoperation and other approaches that require deliberate effort for data collection. EgoDex suggests the human hand as a common embodiment, unlike teleoperation and other approaches that collect data that is only compatible with specific robot hardware platforms.

- EgoDex has 30 FPS 1080p egocentric video with a wide field of view, capturing much of what a human sees while manipulating objects. It has precise and highly detailed 3D pose information for the user's head, arms, wrists, and each joint of each finger from on-device SLAM and calibrated cameras, containing critical dexterous manipulation data unlike in-the-wild Internet videos and Ego4D (Grauman et al., 2022).

- EgoDex consists of extremely diverse behaviors beyond simple pick-and-place such as unscrewing a bottle cap, flipping pages of a book, and plugging a charger into a socket. It consists entirely of active manipulation, unlike existing large egocentric video datasets such as Ego4D.

We systematically evaluate imitation learning policies for hand trajectory prediction to assess the state of the art and identify challenges for future research. We hope that EgoDex will not only accelerate progress in robot manipulation but also be useful more broadly in applications such as computer vision, video generation, and world modeling.

# 2 RELATED WORK

## 2.1 LARGE-SCALE MANIPULATION DATASETS

Several prior works introduce large-scale open-source robot teleoperation datasets, including RoboTurk (Mandlekar et al., 2019), BridgeData (Walke et al., 2023), RT-X (Padalkar et al., 2024), and DROID (Khazatsky et al., 2024). While such datasets contain up to hundreds of hours of valuable

| Dataset | # Traj. | # Tasks | # Frames | Lang. Annot. | Cam. Ext. | Dexterous Annot. | Collection Method |
|---|---|---|---|---|---|---|---|
| **RoboTurk** (Mandlekar et al., 2019) | 2k | 3 | 12M | ✗ | ✗ | ✗ | teleoperation |
| **RoboNet** (Dasari et al., 2019) | 162k | n/a | 15M | ✗ | ✗ | ✗ | scripted |
| **BridgeData V2** (Walke et al., 2023) | 60k | 13 | 2M | ✓ | ✗ | ✗ | teleop+scripted |
| **DROID** (Khazatsky et al., 2024) | 76k | 86 | 19M | ✓ | ✓ | ✗ | teleoperation |
| **EgoMimic** (Kareer et al., 2024) | 2k | 3 | 0.4M | ✗ | ✓ | ✗ | egocentric video |
| **EPIC-KITCHENS** (Damen et al., 2018) | 40k | 125 | 12M | ✓ | ✗ | ✗ | egocentric video |
| **HOI4D** (Liu et al., 2022) | 4k | 54 | 2M | ✗ | ✗ | ✓ | egocentric video |
| **Ego4D (HOI)** (Grauman et al., 2022) | 89k | n/a | 21M | ✓ | ✗ | ✗ | egocentric video |
| **EgoDex (ours)** | **338k** | **194** | **90M** | ✓ | ✓ | ✓ | egocentric video |

Table 1: Comparison of different robot manipulation datasets (above the middle line) and human manipulation datasets (below the middle line). Ego4D (HOI) considers the subset of Ego4D that involves hand-object interaction. EgoDex has the largest amount of trajectories, tasks, and frames by a large margin and has language annotation, camera extrinsics, and dexterous annotation. "Dexterous annotation" is defined here as labels for multi-finger hand poses, which do not include lower fidelity pose data like parallel jaw robot grippers or wrist-only tracking.

manipulation data, it is not clear how to scale the paradigm further than its present scale. Robot teleoperation is extremely labor-intensive and resource-constrained, requiring an operational physical robot and a human teleoperator actively controlling the robot to perform each desired task. Furthermore, it is not clear to what degree such datasets can generalize beyond the set of hardware embodiments and camera viewpoints with which they were collected, even when the datasets consist of samples collected across multiple different embodiments.

Other large-scale datasets such as Ego4D (Grauman et al., 2022) and EPIC-KITCHENS (Damen et al., 2018) consist of egocentric video recording humans performing various activities. While such datasets are more scalable and not limited to particular hardware platforms, they typically do not focus on manipulation and do not have paired 3D annotations for dexterous manipulation.

There is also a large body of work that considers hand-object interaction (Liu et al., 2022; Banerjee et al., 2025; Chao et al., 2021; Zhou et al.). While these datasets often include 3D hand pose annotations, they are orders of magnitude smaller than EgoDex due to manual annotation. Moreover, their emphasis is primarily on grasping rather than diverse and long-horizon manipulation tasks.

## 2.2 Scalable Methods for Robot Data Collection

Recent work identifies the data scarcity problem in robot imitation learning and proposes innovative techniques for scalable data collection. Chi et al. (2024) propose the "universal manipulation interface": handheld grippers that enable human teachers to provide demonstrations without physical robots. Wang et al. (2024) introduces a portable data collection system with motion capture gloves. Others propose collecting robot-free demonstrations by simulating robot hardware in augmented reality (Chen et al., 2024b; Park et al., 2024; Nechyporenko et al., 2024).

These approaches all face a similar pitfall: they require *active* data collection. While they may make it easier to collect data than teleoperation, human demonstrators must still be incentivized to intentionally collect the data. Such approaches face a significant uphill battle in approaching the scale of Internet datasets, where text and images are not deliberately collected but rather a passive byproduct of human interaction with the Internet.

## 2.3 Learning from Human Video

Video data is abundant on the Internet. Prior work explores representation learning on unstructured large-scale image and video data for pretraining visual encoders (Radosavovic et al., 2023; Ma et al., 2023) and grasp affordances (Bahl et al., 2023) for downstream manipulation. However, raw unstructured video data faces a prohibitively large gap between its image distribution and that of a dexterous manipulation task. Moreover, such videos are not labeled with corresponding motor actions with which to train a policy.

One option is to postprocess the unstructured video data with 3D hand prediction networks such as HaMeR (Pavlakos et al., 2024), recently explored by Ren et al. (2025). However, the prediction

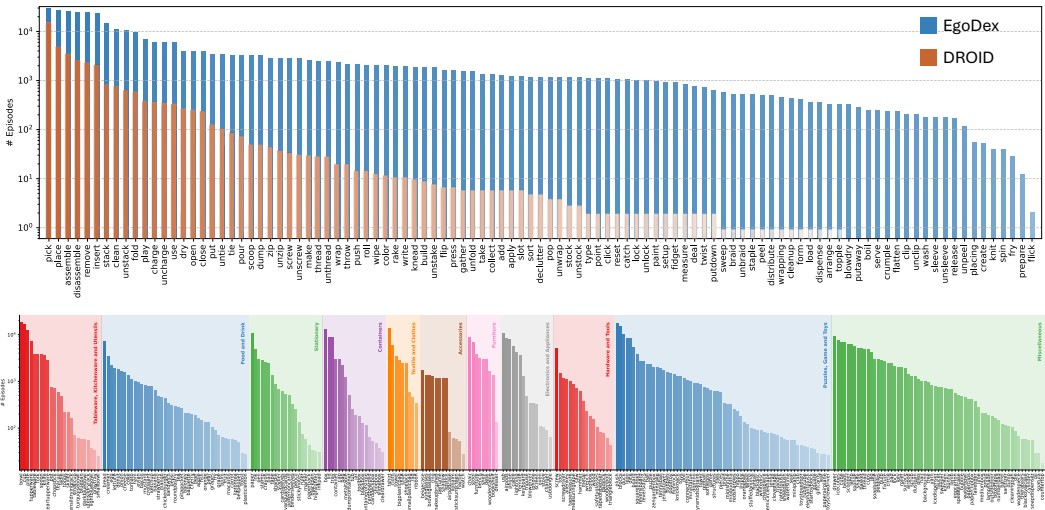

Figure 2: Distribution of EgoDex dataset. **Top**: Distribution of distinct verbs, sorted by frequency. The horizontal axis is verbs of EgoDex. The orange plot is taken from DROID (Khazatsky et al., 2024). While many verbs in DROID are below the $10^1$ mark, most verbs in EgoDex are above the $10^3$ mark. **Bottom**: Distribution of distinct objects. The clustering is suggested by GPT-4.

quality of these networks can suffer without multiple viewpoints and detailed knowledge of the camera extrinsics at all times, usually unavailable with raw Internet video. In contrast, the EgoDex dataset includes 3D head and hand tracking *at the time of collection*, where multiple cameras on the Vision Pro, known intrinsics and extrinsics, and a production-grade hand prediction network all contribute to precise annotation.

Most similar to our work is EgoMimic (Kareer et al., 2024), which proposes the collection of ego-centric video and paired 3D hand tracking. The primary difference is scale: while EgoMimic collects around 4 hours of data, we collect 829 hours with a much broader data and task distribution. We also collect more dexterous annotations, critical for downstream manipulation: 3D positions and orientations for the upper body including the head, shoulders, arms, and 25 joints in each hand, whereas EgoMimic collects only the wrist positions.

## 3 EGODEX DATASET

The EgoDex dataset contains 829 hours of 1080p, 30 Hz egocentric video with 338000 episodes across 194 tasks. This is a total of 90 million frames. The dataset also has rich and structured annotations including natural language, camera extrinsics, and hand pose annotations (Section 3.2). The full dataset takes 2.0 TB of storage on disk. We compare EgoDex to existing manipulation datasets in Table 1.

### 3.1 DATA COLLECTION

All data is collected with Apple Vision Pro running visionOS 2. The high-resolution and high-frequency passthrough and wide field of view enable intuitive egocentric data collection, where the collector can observe the environment unobstructed as if with their own eyes, and the camera data records precisely what the collector sees without any pose offsets (unlike, for instance, a head-mounted camera). We use ARKit, a production-grade pose tracking software, to collect natural demonstrations with bare hands and without any additional hardware apparatus.

To streamline data collection, data is recorded in *sessions*: approximately 10-15 minute segments that consist of many individual episodes, where episode boundaries are indicated by a "pause" and subsequent "resume" of recording from the data collection app. Raw video is compressed to facilitate data transfer, upload, download, and storage. Without the use of modern video compression

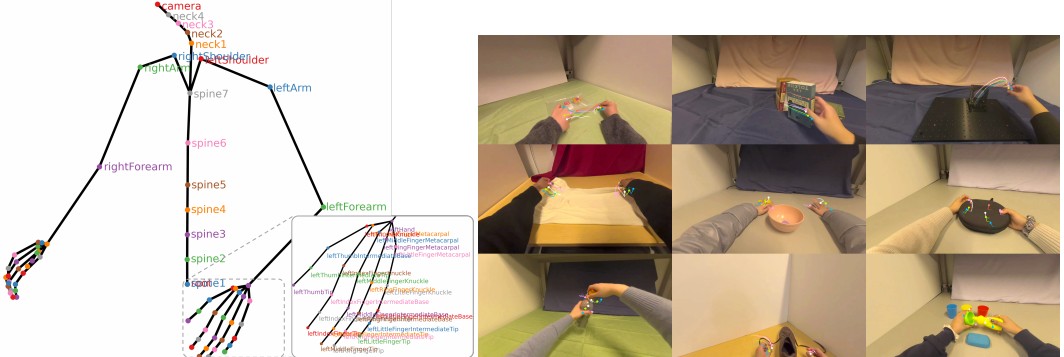

Figure 3: **Left**: Joints captured by EgoDex. **Right**: Examples of dexterous manipulation behaviors. Tracked fingertips are highlighted in distinct colors and show 0.5 seconds of motion before the current frame. From left to right, top to bottom, the tasks are: unzipping a Ziploc bag, removing a book from a bookshelf, removing a screw from a fixture, folding a t-shirt, decluttering, opening a case, unscrewing a bottle cap, tying shoelaces, and washing a cup.

algorithms, the raw data would take over 500 TB of disk space, about 250× its current size. At training time, data is loaded efficiently with PyTorch torchcodec (Team, 2024), which only decodes the desired frames in the sampled batch of data.

## 3.2 MODALITIES

The data consists of the following: 1) Egocentric RGB video with 1920 × 1080 resolution at 30 Hz frequency. 2) Camera intrinsics and extrinsics at 30 Hz. 3) 3D position and orientation of all upper body joints (including 25 joints for each hand) at 30 Hz. 4) Confidence values for pose predictions at 30 Hz. 5) Natural language annotation of the manipulation.

The metadata annotated by data collectors includes the task name, a brief task description in natural language, details about the environment, and details about the object(s) that are manipulated. Since the metadata can be noisy, these fields are provided as input to GPT-4 (OpenAI et al., 2024), which combines this information into a single detailed natural language description.

Confidence values are scalars between 0 and 1, indicating the ARKit prediction confidence per skeletal joint. A confidence of zero indicates that the joint is fully occluded from view. See Appendix A.3 for a comprehensive list of all the joints and more information.

## 3.3 TASK TYPES

EgoDex consists of 3 types of tasks:

- *Reversible* tasks are pairs of tasks that are the inverse of each other. The distribution of final states for one task is within the distribution of initial states for its inverse. For example, connecting a charger to a device and removing a charger from the device.
- *Reset-free* tasks are tasks with a final state distribution that falls within its *own* initial state distribution. For example, throwing a ball in the air and catching it (where gravity acts as the reset).
- *Reset* tasks are tasks in which the environment must be reset to the initial state distribution after each demonstration.

Reversible and reset-free tasks enable a higher yield from data collection as they eliminate costly resets, which are not included in the recorded data.

## 3.4 DIVERSITY

Prior works (Khazatsky et al., 2024; Padalkar et al., 2024) identify several potential axes of demonstration diversity: viewpoint diversity, task diversity, scene diversity, object diversity, and more. In

EgoDex, the emphasis is on diversity in *dexterous manipulation behaviors*. Tasks and objects vary such that the required dexterity ranges far beyond pick-and-place, the primary behavior in most robot teleoperation datasets. For example, tasks include tightening a screw, tying shoelaces, dealing cards, flipping pages, catching tennis balls, and slotting batteries. The task distribution covers a wide range of everyday household manipulation tasks that can be performed on a tabletop surface. There is also a significant amount of basic pick-and-place with diverse objects, as well as the benchmark tasks from the FurnitureBench assembly benchmark (Heo et al., 2023). The full list of 194 tasks is provided in Appendix A.2.

To get a sense of the spread of the task distribution as in prior work, we plot the distribution of deduplicated verbs in Figure 2. We observe that the distribution is much wider than prior works such as DROID (Khazatsky et al., 2024), where a large fraction of verbs have less than $10^1$ demonstrations and sometimes only a single demonstration; in contrast, most of the verbs in EgoDex have more than $10^3$ demonstrations.

Still, the verb distribution does not capture the full diversity of manipulation behaviors or tasks. For example, "assemble" can involve radically different behaviors in the context of different objects and tasks. See Figure 3 for examples of different dexterous manipulation behaviors captured in the dataset.

While the *scene* diversity in EgoDex is limited to tabletop environments, the Cartesian product of scene and behavior is not the focus of our work, which focuses on behavioral diversity. Scene diversity can be introduced with modern visual data augmentation methods such as image-to-image generative models (Yuan et al., 2025; Chen et al., 2024a).

## 4 EGODEX BENCHMARKS

### 4.1 ACTION REPRESENTATION

Given the full set of skeletal joints in the EgoDex dataset, many action representations are possible: wrist positions, wrist orientation, positions of fingertips, and so on. Since we focus on dexterous manipulation, we choose a representation that captures sufficient bimanual dexterity. Specifically, the action $\mathbf{a_t}$ at time $t$ is represented as the 3D position of each wrist, the 6D orientation of each wrist (as suggested by Zhou et al. (2018)), and the 3D position of each fingertip. Thus, each action has a total dimensionality of 2 hands $\times$ (3 + 6 + (3 $\times$ 5 fingertips)) = 48. In practice, actions are predicted in chunks over a fixed time horizon. Poses are expressed in the current camera frame (Kareer et al., 2024), and each action chunk is a relative trajectory (Chi et al., 2024).

### 4.2 BENCHMARKS

We propose two benchmark tasks for EgoDex. The first is *dexterous trajectory prediction*: from the egocentric image observations, skeletal joint poses, and natural language description, the task is to predict the trajectories of the hands for a given time horizon following the observations. Specifically, we seek to train the following estimator:

$$f_\theta(\mathbf{o}_{0..t}, \mathbf{s}_{0..t}, l) = \hat{\mathbf{a}}_{t:t+H}$$

where $\mathbf{o}_{0..t}$ are the egocentric image observations up to and including time $t$, $\mathbf{s}_{0..t}$ are skeletal pose observations up to and including time $t$, $l$ is a natural language description, $\hat{\mathbf{a}}_{t:t+H}$ is the predicted action chunk, and $H$ is the prediction horizon.

Since multimodality can be very severe in natural human motion, the second benchmark is *inverse dynamics*: from the image observations and skeletal poses up to time $t$ as well as a goal image observation at the end of the time horizon, the task is to predict the trajectories of the hands in between the start and end observations. In this case, we train the following estimator, which can be interpreted as a visually goal-conditioned policy:

$$f_\theta(\mathbf{o}_{0..t}, \mathbf{s}_{0..t}, \mathbf{o}_{t+H}, l) = \hat{\mathbf{a}}_{t:t+H}$$

Each of these benchmarks are parameterized by prediction horizon $H$. For example, a short-horizon trajectory prediction task may set $H = 30$ (1 second), while a more difficult long-horizon task may set $H = 90$ (3 seconds).

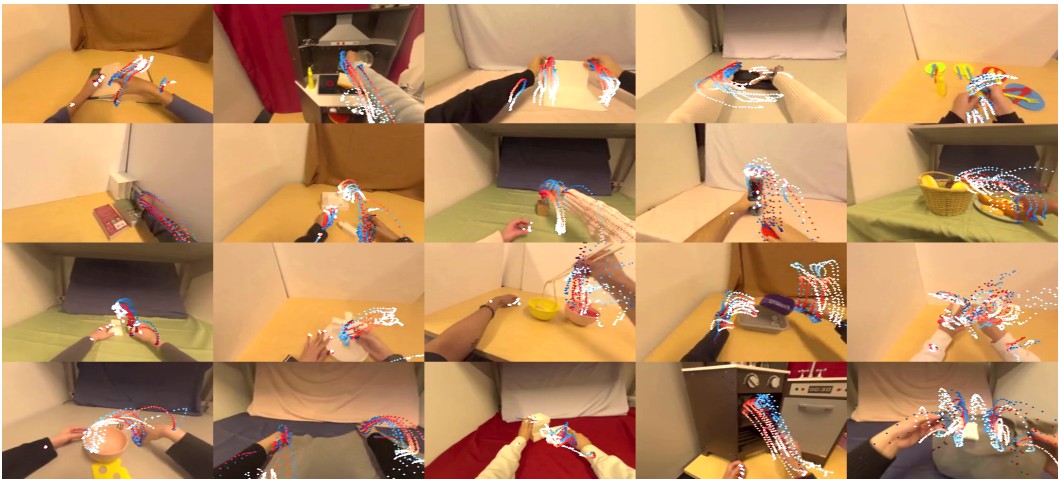

Figure 4: Model prediction visualizations for Dec + BC on test set images with a 2 second horizon. Blue trajectories are ground truth and red trajectories are predictions, where darker colors and closer to the current frame and lighter colors are further in the future. The points shown are the wrist and fingertip positions projected into the camera frame (a total of 12 trajectories).

Unlike typical robot hardware experiments that can vary across physical environments, the EgoDex benchmarks are fully reproducible with a fixed training and test set. We set aside 1% of the EgoDex dataset as a fixed held-out test set for evaluations, where the remaining 99% can be split across training and validation as desired.

## 4.3 EVALUATION METRICS

Since trajectory prediction for natural human motion is inherently multimodal, evaluating a single predicted trajectory against the ground truth sample may be insufficient for measuring correctness. For example, for the simple task of placing a fruit in a basket, it could be placed at variable locations within the basket, moved at variable speeds, and moved in different but equally valid trajectories from the initial position to the basket.

Thus, for each benchmark task we evaluate performance with a "best of $K$" metric. For each data point in the full test set, we sample the trained model $K$ times to capture different possible modes. We then compute the distance between the ground truth trajectory and the trajectory closest to it out of the $K$ samples, where "distance" is calculated as the Euclidean distance between predicted 3D keypoint positions and their ground truth 3D counterparts, averaged over each timestep in the predicted chunk and each of the 12 keypoints (i.e., the wrist and fingertips of each hand). Intuitively, this value can be interpreted as the average positional error in 3D space between ground truth and prediction in meters. The final value is averaged over the full test set. For deterministic models, the value is the same regardless of $K$; for stochastic models, the value improves as $K$ increases, as the model gets more chances to sample the ground truth mode.

## 5 EXPERIMENTS

We train and evaluate state-of-the-art imitation learning policies from the X-IL framework (Jia et al., 2025) on the benchmarks from Section 4. Specifically, we train two Transformer model architectures (encoder-decoder and decoder-only) and three policy representations (behavior cloning, denoising diffusion, and flow matching). We also run experiments to evaluate the effect of prediction horizon, visual goal-conditioning, dataset size, and model size. In total we train and evaluate 14 different models. We train all models for 50,000 gradient steps with a batch size of 2048 parallelized across 8 NVIDIA A100 GPUs. To make the train-test split, we randomly sample a 1% subset of each task and set it aside as a held-out set for evaluation. Since this test set does not contain out-of-distribution (OOD) tasks, we provide additional OOD experiments in Appendix A.1. Additional training and

| Model | Avg Distance (m) | | | Final Distance (m) | | |
|---|---|---|---|---|---|---|
| | $K = 1$ | $K = 5$ | $K = 10$ | $K = 1$ | $K = 5$ | $K = 10$ |
| Dec + BC | 0.045 | 0.045 | 0.045 | 0.062 | 0.062 | 0.062 |
| Dec + DDPM | 0.053 | 0.044 | 0.041 | 0.071 | 0.050 | 0.044 |
| Dec + FM | 0.052 | 0.042 | 0.040 | 0.071 | 0.049 | 0.043 |
| EncDec + BC | **0.044** | 0.044 | 0.044 | **0.060** | 0.060 | 0.060 |
| EncDec + DDPM | 0.052 | 0.042 | 0.039 | 0.071 | 0.048 | 0.043 |
| EncDec + FM | 0.051 | **0.041** | **0.038** | 0.070 | **0.047** | **0.041** |

Table 2: Evaluations for different models on trajectory prediction with a 2 second horizon.

| Model | Avg Distance (m) | | | Final Distance (m) | | |
|---|---|---|---|---|---|---|
| | $H = 30$ (1s) | $H = 60$ (2s) | $H = 90$ (3s) | $H = 30$ (1s) | $H = 60$ (2s) | $H = 90$ (3s) |
| Dec + BC | **0.031** | 0.045 | 0.053 | **0.049** | 0.062 | 0.069 |

Table 3: Results for models trained and evaluated with different prediction horizons. As expected, accuracy falls as the prediction horizon increases. $H = 60$ values are repeated from Table 2 for convenience.

model details are provided in Appendix A.4. The results are presented in Tables 2, 3, 4 and Figure 5 and summarized below.

**Encoder-decoder architectures outperform decoder-only.** In Table 2 we observe that all encoder-decoder ("EncDec") models consistently outperform their decoder-only ("Dec") counterparts by a small margin.

**Different policy representations excel in different settings.** In Table 2 we observe that the encoder-decoder flow matching ("FM") model outperforms the other models for $K = 5$ and $K = 10$ by up to 34%. As expected, denoising diffusion ("DDPM") and FM evaluations improve as $K$ increases, while behavior cloning ("BC") remains the same independent of $K$ as it is deterministic. Note however that for the $K = 1$ setting, BC outperforms both diffusion and flow-matching by about 15%. This suggests that the average prediction of BC is better than DDPM and FM, while the best prediction of DDPM and FM is better than BC's single prediction.

**Performance degrades as the prediction horizon increases.** In the remaining experiments we vary different properties while fixing the model to the simplest policy: decoder-only behavior cloning. In Table 3 we see that reducing the horizon from 2 seconds to 1 second improves average and final distance by 31% and 21% respectively, while increasing the horizon from 2 to 3 seconds worsens average and final distance by 18% and 11% respectively. Intuitively, accurate prediction becomes more challenging as the horizon increases as the model must predict 48-dimensional dexterous actions farther into the future. Appendix A.1 shows additional experiments using the encoder-decoder with flow matching (EncDec + FM) model, which shows a similar trend.

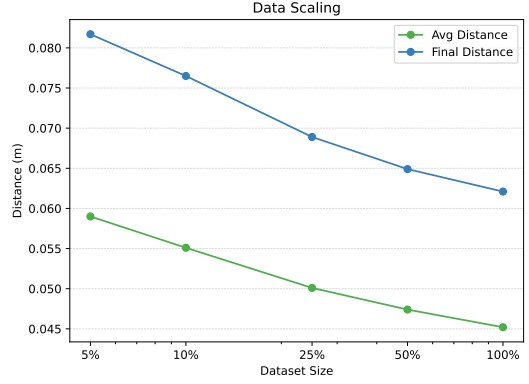

Figure 5: Distance metrics w.r.t. training dataset size, where size is plotted on a log-scale. Performance improves as the dataset gets larger.

**Visual goal-conditioning significantly improves performance.** In Table 4 we observe that visual goal-conditioning reduces average distance by 22% and final distance by 53%. Intuitively, a visual goal provides a visual "anchor" to ground the endpoint of the predicted trajectory and mitigate multimodality. This yields a baseline score for the inverse dynamics benchmark specified in Section 4.

| Model | Avg Distance (m) | Final Distance (m) |
|---|---|---|
| Dec + BC | 0.045 | 0.062 |
| Dec + BC w/ goal image | **0.035** | **0.029** |

Table 4: Visual goal-conditioning results. Training a model with visual goal conditioning reduces average distance by 22% and final distance by 53%.

**Medium-size model capacity is sufficient for the current dataset size.** We train and evaluate a larger Dec+BC model with 500 million parameters as opposed to the default 200 million parameters. The larger model attains average distance 0.045 and final distance 0.062, exactly the same as the default 200 million parameter model. This may increase accessibility for the EgoDex benchmarks, as medium-size models fit comfortably on commodity GPU hardware.

**Performance scales with dataset size.** In Figure 5 we observe that average and final distance improve as dataset size increases. Results suggest that performance scales with data, motivating the collection of large-scale egocentric datasets like EgoDex.

## 6 RESEARCH USE CASES

**Robotics** While significant progress has been made in the development of robot hardware with humanoid morphologies and dexterous hands, there remains a prohibitive embodiment gap between humans and today's robots. Some options for bridging the embodiment gap include 1) co-training with a small-scale robot dataset, as demonstrated by Kareer et al. (2024); Qiu et al. (2025); 2) pretraining with large-scale human data and supervised fine-tuning with smaller-scale robot data, similar to the training recipe for large language models; 3) training a visual encoder on the human data for more data-efficient imitation learning downstream (Nair et al., 2022); 4) learning robot manipulation priors from the human-object interaction trajectories and then fine-tuning with reinforcement learning or imitation learning (Singh et al., 2024; Gavryushin et al., 2025).

**Perception** EgoDex can be used for learning tasks such as action recognition and human-object interaction detection. Datasets like EPIC-KITCHENS (Damen et al., 2018) have demonstrated the value of egocentric video for recognizing and anticipating daily actions, and even more challenging tasks like detecting active objects and predicting state changes from egocentric video. Researchers can also study which objects are involved in each action and how. For example, one could model the contact points, grasps, and trajectories when using a tool (screwdriver, scissors, etc.). A related task is learning object affordances, i.e., understanding what actions each object supports.

**Video Generation and World Models** Recent advances in large-scale diffusion models have significantly enhanced the capabilities of language-conditioned video generation, producing temporally consistent and semantically accurate visual narratives from natural language inputs (Li et al., 2024; Peng et al., 2025; NVIDIA et al., 2025; Zhao et al., 2025). These generative frameworks have demonstrated potential not only in creating realistic and detailed video content but also as world models for decision-making tasks, supporting reinforcement learning agents by simulating future outcomes based on predicted visual dynamics (Wang et al., 2025; Bruce et al., 2024; Yang et al., 2023; Hafner et al., 2019). Despite these impressive advancements, there remains a substantial research gap in video generation and world modeling from an egocentric viewpoint. Egocentric perspectives present unique challenges, including managing significant viewpoint variability, maintaining temporal and spatial consistency amid frequent camera movements, and accurately reflecting agent-centric interactions and intentions. Since EgoDex provides large-scale video data with 3D pose and language annotations, it enables the training of an egocentric world model.

## 7 CONCLUSION

We introduce EgoDex, a massive dataset of egocentric video paired with 3D pose annotations in a wide range of dexterous manipulation tasks. We train and evaluate imitation learning policies for hand trajectory prediction on this data.

While EgoDex has significant diversity across tasks and manipulation behaviors, it is limited in background and scene diversity. The dexterous annotations can also be imperfect, especially during heavy occlusion (e.g., towel folding) or very high speed motions, as they are themselves model predictions. Future work involves procedural background randomization on the existing data (Yuan et al., 2025) as well as data collection in more diverse environments.

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

## A APPENDIX

### A.1 ADDITIONAL EXPERIMENTS

The EgoDex dataset also includes a set of 6 entirely out-of-distribution (OOD) tasks in the dataset under a separate folder (titled extra). We ran additional experiments testing the decoder-only behavior cloning (Dec+BC) model on these OOD tasks. We observe that some OOD tasks are comparable to in-distribution performance, while tasks that are further out of distribution have worse performance. This suggests EgoDex models can generalize to OOD tasks that are at least somewhat similar to in-distribution tasks.

| Model | Avg Distance (m) | Final Distance (m) |
|---|---|---|
| In-Distribution Avg | 0.045 | 0.062 |
| (OOD) Jigsaw Puzzle | 0.047 | 0.065 |
| (OOD) Tetra Board | 0.060 | 0.082 |
| (OOD) Knit Scarf | 0.064 | 0.093 |
| (OOD) Play Reversi | 0.065 | 0.096 |
| (OOD) Blowdry Hair | 0.083 | 0.118 |
| (OOD) Stamp Paper | 0.099 | 0.162 |

Table 5: Additional experimental results on out-of-distribution tasks.

In main experiments, we chose to use a decoder-only BC model to evaluate the effect of increasing the prediction horizon. We additionally ran experiments to evaluate the effect of the prediction horizon with the encoder-decoder with flow matching (EncDec+FM) model for further clarity. The trend is consistent with the Dec+BC model, where performance degrades as the horizon increases.

| | Avg Distance (m) | | | Final Distance (m) | | |
|---|---|---|---|---|---|---|
| Model | $K = 1$ | $K = 5$ | $K = 10$ | $K = 1$ | $K = 5$ | $K = 10$ |
| H = 30 (1s) | 0.036 | 0.028 | 0.026 | 0.055 | 0.037 | 0.033 |
| H = 60 (2s) | 0.051 | 0.041 | 0.038 | 0.070 | 0.047 | 0.041 |
| H = 90 (3s) | 0.061 | 0.050 | 0.047 | 0.079 | 0.053 | 0.046 |

Table 6: Ablation of prediction horizon for the encoder-decoder with flow matching (EncDec+FM) model.

### A.2 COMPLETE LIST OF TASKS

We provide a complete list of task names here, labeled as they appear in the dataset and separated by task type (reversible, reset-free, or reset, with definitions in Section 3.3). Recall that each reversible task is actually a pair of two tasks. There are a total of 194 tasks. See Figure 6 for a visual of a subset of the objects used in the various manipulation tasks.

For users interested in robot deployment, the `basic_pick_place` task in particular has a large amount of very diverse pick-and-place data as well as high-quality language annotation.

**Reversible (76 × 2 total tasks):**

- `braid_unbraid`
- `charge_uncharge_airpods`
- `deal_gather_cards`
- `fry_bread`
- `assemble_disassemble_furniture_bench_chair`
- `assemble_disassemble_furniture_bench_drawer`
- `assemble_disassemble_furniture_bench_square_table`

- `fold_unfold_paper_basic`
- `insert_remove_furniture_bench_cabinet`
- `gather_roll_dice`
- `insert_remove_airpods`
- `insert_remove_drawer`
- `insert_remove_shirt_in_tube`
- `insert_remove_usb`
- `load_dispense_ice`
- `open_close_insert_remove_tupperware`
- `pick_up_and_put_down_case_or_bag`
- `put_away_set_up_board_game`
- `screw_unscrew_fingers_fixture`
- `sleeve_unsleeve_cards`
- `stack_unstack_cups`
- `thread_unthread_bead_necklace`
- `tie_and_untie_shoelace`
- `insert_remove_tennis_ball`
- `open_close_insert_remove_case`
- `pick_place_food`
- `put_in_take_out_glasses`
- `screw_unscrew_allen_fixture`
- `set_up_clean_up_chessboard`
- `slot_batteries`
- `stack_unstack_bowls`
- `stack_unstack_tupperware`
- `throw_collect_objects`
- `vertical_pick_place`
- `wash_put_away_dishes`
- `add_remove_lid`
- `arrange_topple_dominoes`
- `assemble_disassemble_legos`
- `assemble_disassemble_soft_legos`
- `assemble_disassemble_structures`
- `assemble_disassemble_tiles`
- `boil_serve_egg`
- `build_unstack_lego`
- `charge_uncharge_device`
- `clip_unclip_papers`
- `crumple_flatten_paper`
- `fry_egg`
- `assemble_disassemble_furniture_bench_desk`
- `assemble_disassemble_furniture_bench_lamp`
- `assemble_disassemble_furniture_bench_stool`
- `fold_stack_unstack_unfold_cloths`

- `fold_unfold_paper_origami`
- `insert_remove_furniture_bench_round_table`
- `insert_remove_bagging`
- `insert_remove_cups_from_rack`
- `insert_remove_plug_socket`
- `insert_remove_utensils`
- `lock_unlock_key`
- `open_close_insert_remove_box`
- `scoop_dump_ice`
- `screw_unscrew_bottle_cap`
- `setup_cleanup_table`
- `stock_unstock_fridge`
- `stack_unstack_plates`
- `throw_and_catch_ball`
- `tie_untie_rubberband`
- `wrap_unwrap_food`
- `zip_unzip_bag`
- `zip_unzip_case`
- `assemble_disassemble_jigsaw_puzzle`
- `stack_unstack_tetra_board`
- `stack_remove_jenga`
- `insert_dump_blocks`
- `rake_smooth_zen_garden`
- `play_reset_connect_four`
- `insert_remove_bookshelf`

**Reset-free (28 total tasks):**

- `color`
- `fidget_magnetic_spinner_rings`
- `measure_objects`
- `staple_paper`
- `use_rubiks_cube`
- `wash_kitchen_dishes`
- `wipe_screen`
- `knead_slime`
- `point_and_click_remote`
- `type_keyboard`
- `clean_surface`
- `dry_hands`
- `play_mancala`
- `flip_coin`
- `flip_pages`
- `paint_clean_brush`
- `play_piano`

- push_pop_toy
- put_toothpaste_on_toothbrush
- wash_fruit
- wipe_kitchen_surfaces
- stamp_paper
- blowdry_hair
- knit_scarf
- makeup
- write
- clean_cups
- roll_ball

**Reset (14 total tasks):**

- clean_tableware
- declutter_desk
- basic_pick_place
- stack
- make_sandwich
- peel_place_sticker
- sweep_dustpan
- wrap
- assemble_jenga
- basic_fold
- pour
- sort_beads
- use_chopsticks
- play_reversi

### A.3 COMPLETE LIST OF SKELETAL JOINTS

The annotations consist of SE(3) poses (represented as $4 \times 4$ homogeneous transformation matrices) for each of the following joints, labeled by their names as they appear in the dataset:

**Upper Body:**

```
hip, spine1, spine2, spine3, spine4, spine5, spine6, spine7,
neck1, neck2, neck3, neck4, leftShoulder, leftArm, leftForearm,
leftHand, rightShoulder, rightArm, rightForearm, rightHand
```

**Left Hand:**

```
leftIndexFingerIntermediateBase, leftIndexFingerIntermediateTip,
leftIndexFingerKnuckle, leftIndexFingerMetacarpal,
leftIndexFingerTip, leftLittleFingerIntermediateBase,
leftLittleFingerIntermediateTip, leftLittleFingerKnuckle,
leftLittleFingerMetacarpal, leftLittleFingerTip,
leftMiddleFingerIntermediateBase, leftMiddleFingerIntermediateTip,
leftMiddleFingerKnuckle, leftMiddleFingerMetacarpal,
leftMiddleFingerTip, leftRingFingerIntermediateBase,
leftRingFingerIntermediateTip, leftRingFingerKnuckle,
leftRingFingerMetacarpal, leftRingFingerTip,
```

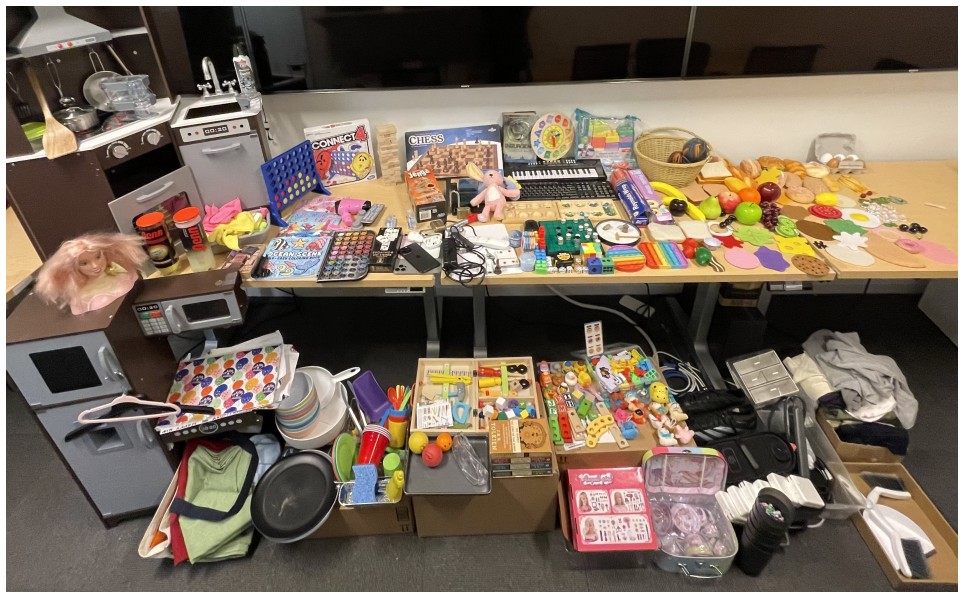

Figure 6: Some of the objects used in the various manipulation tasks.

```
leftThumbIntermediateBase, leftThumbIntermediateTip,
leftThumbKnuckle, leftThumbTip
```

**Right Hand:**

```
rightIndexFingerIntermediateBase, rightIndexFingerIntermediateTip,
rightIndexFingerKnuckle, rightIndexFingerMetacarpal,
rightIndexFingerTip, rightLittleFingerIntermediateBase,
rightLittleFingerIntermediateTip, rightLittleFingerKnuckle,
rightLittleFingerMetacarpal, rightLittleFingerTip,
rightMiddleFingerIntermediateBase,
rightMiddleFingerIntermediateTip, rightMiddleFingerKnuckle,
rightMiddleFingerMetacarpal, rightMiddleFingerTip,
rightRingFingerIntermediateBase, rightRingFingerIntermediateTip,
rightRingFingerKnuckle, rightRingFingerMetacarpal,
rightRingFingerTip, rightThumbIntermediateBase,
rightThumbIntermediateTip, rightThumbKnuckle, rightThumbTip
```

Note that `leftHand` and `rightHand` refer to the wrists. Note also that the joint confidence values in the data behave differently for the wrists and the hands. Wrist confidence values (for `leftHand` and `rightHand`) indicate whether each hand is detected as a whole, while finger joint confidence values indicate confidence *relative* to the wrist. If, for instance, the left index fingertip has high confidence but the left wrist has low confidence, it is unlikely that the left index fingertip is reliable.

### A.4 TRAINING DETAILS

In the experiments section we train and evaluate 14 different models: 6 combinations of architectures and policy optimization methods, 4 additional models with different training dataset sizes, 2 additional models with different prediction horizons, 1 model with a larger model size, and 1 model with visual goal-conditioning. See Figure 7 for intuition on the model architecture.

Each model is trained and evaluated on a single node with 96 logical CPUs (48 physical CPUs) and 8 NVIDIA A100 GPUs each with 80GB RAM. Training is run for 50,000 gradient steps with a batch size of 2048 (256 per GPU with data parallelism), at which point training and validation loss plateau. The full training run takes approximately 72 hours. The models are optimized with Adam and a learning rate of 1e-4. Image observations are resized to $224 \times 224$ and sent through a pretrained ResNet encoder, while language annotations are passed through a frozen CLIP (Radford

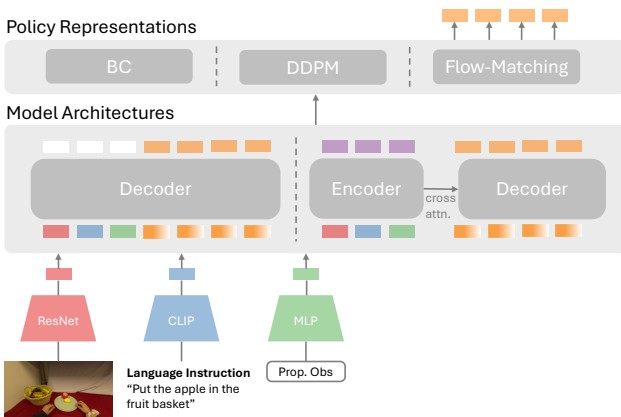

Figure 7: Model architectures.

et al., 2021) encoder. Only the current image observation and proprioceptive state are passed as input to the policy (i.e., no history); adding history may improve performance. DDPM and FM models are trained and evaluated with 16 sampling steps. All other hyperparameters are the defaults from the X-IL codebase (Jia et al., 2025).

