# OpenReview forum: "EgoDex: Learning Dexterous Manipulation from Large-Scale Egocentric Video"
_ICLR.cc/2026/Conference — ICLR 2026 Poster_

### Official Review · Reviewer_7T6A · 2025-10-30

**Soundness:** 2
**Presentation:** 3
**Contribution:** 2
**Rating:** 4
**Confidence:** 3

**Summary:**

This paper presents a large-scale egocentric human video dataset that captures diverse manipulation behaviors in everyday scenarios, annotated with 3D hand poses and tracking information. To demonstrate the dataset’s utility for robot manipulation learning, an imitation learning policy is trained to predict hand trajectories, and a set of benchmark tasks and evaluation metrics are introduced.

**Strengths:**

Leveraging human videos to learn dexterous manipulation policies is a promising direction, especially since collecting teleoperation data is often costly and time-consuming.

The authors’ plan to open-source the dataset will further benefit the research community and foster future developments in this area.

**Weaknesses:**

The evaluation is limited, focusing solely on the trajectory prediction task.

The paper lacks experiments demonstrating that progress on trajectory prediction tasks can effectively transfer to real-world robotic manipulation. Additional empirical studies validating this transferability would further strengthen the work and better justify the practical usefulness of the proposed dataset.

Moreover, based on the provided sample data, the captured human behaviors appear somewhat unnatural---for instance, frequent use of pinch grasps---and the backgrounds are relatively clean and uncluttered, which may limit the dataset’s realism and representativeness of everyday human activities.

**Questions:**

What are the failure modes of the learned models? Could the authors provide more qualitative results to illustrate these cases?

Beyond the trajectory prediction task, do the authors plan to include additional evaluation tasks to further assess the dataset’s utility?

---

> ### Author Response · Authors · 2025-11-21
>
> Thanks to the reviewer for the constructive feedback! We appreciate that the reviewer acknowledges the promising research directions and benefits to the broader research community. Here is our response to address the remaining questions and concerns:
>
> 1. Limited evaluation:
>
> We acknowledge that success in human motion prediction does not necessarily translate to real-world robotic manipulation success rate. However, the primary objective of this work is to unblock the community by addressing the critical lack of scale and structure in existing human datasets. The downstream integration of this data into control policies is a significant undertaking in its own right, a fact highlighted by several recent works [1–4] dedicated specifically to this challenge. Notably, H-RDT [2] has demonstrated that pre-training on EgoDex alone yields a 40.5% improvement in real-world success rates over SOTA baselines like Pi0 and RDT.
>
> We maintain that our proposed benchmarks (Trajectory Prediction and Inverse Dynamics) provide a rigorous evaluation of representation quality, isolating the learned features from confounding variables such as robot kinematics, hardware specifics, and retargeting noise. We thank the reviewer for pointing out the difference and will clarify the distinction between representation quality and policy transfer in the final manuscript.
>
> [1] R. Qiu et al. Humanoid Policy ~ Human Policy. Conference on Robot Learning (CoRL), 2025.
>
> [2] H. Bi et al. H-RDT: Human Manipulation Enhanced Bimanual Robotic Manipulation. arXiv preprint arXiv:2507.23523, 2025.
>
> [3] H. Luo et al. "Being-h0: vision-language-action pretraining from large-scale human videos." arXiv preprint arXiv:2507.15597 (2025).
>
> [4] Y. Jiang et al. "RynnVLA-001: Using Human Demonstrations to Improve Robot Manipulation." arXiv preprint arXiv:2509.15212 (2025).
>
> 2. Unnatural human behavior and realism of dataset:
>
> We thank the reviewer for this observation regarding the sample data. It is important to note that the provided samples represents only a small subset of our 200-task dataset and do not reflect the full dataset statistics. Pinch grasps are only used in a small number of tasks like "basic_pick_place" to facilitate easier robot gripper transfer. The full collection encompasses a diverse range of manipulation tasks, the vast majority of which require dexterity significantly beyond simple pinch grasps (e.g., braiding/unbraiding yarn, dealing playing cards, and fastening fixtures). We prioritized these complex, free-form manipulations as they offer a more scalable path for data collection compared to highly constrained, precision-based tasks.
>
> Furthermore, while not fully in-the-wild, the dataset captures systematic environmental variation across multiple dimensions including table appearance, background composition, and camera viewpoint. This diversity ensures the data is representative of messy, real-world manipulation scenarios rather than idealized laboratory conditions.
>
> 3. Failure modes:
>
> The main failure mode we observed was from Out-of-Distribution tasks. We analyzed this quantitatively in Appendix A.1 (Table 5) by evaluating on a set of extra tasks that we included as part of the dataset. The model generalizes well to OOD tasks that share kinematic primitives with the training set (e.g., "Jigsaw Puzzle", comparable to in-distribution). However, performance degrades on tasks with highly distinct object dynamics or tool usage not represented in the training data, such as "Blowdry Hair". Another factor that contributes to trajectory prediction error is severe multimodality in natural human data, as suggested by the delta between models trained with visual goal-conditioning and those without (Table 4).
>
> 4. Further assessment of dataset's utility:
>
> As mentioned above, recent works [1–4] that utilized EgoDex have shown the utility of our dataset and how applying this data to robot control policies is complex and a significant contribution in its own right. We maintain that the scope of our paper is on the dataset contribution and our proposed benchmarks on trajectory prediction provide a rigorous evaluation of the data quality, isolating the learned features from confounding variables such as robot kinematics, hardware specifics, and retargeting noise.

---

> > ### Comment · Reviewer_7T6A · 2025-11-25
> >
> > Thanks to the authors for the clarification. After reviewing the response, I believe most of my concerns have been addressed. I still feel that further validation on robotics tasks would strengthen the claims regarding the dataset’s usefulness for bootstrapping robot manipulation policy training. Nevertheless, I plan to raise my score accordingly.

---

> > > ### Author Response · Authors · 2025-11-25
> > >
> > > Thanks for the support! We're glad to see most of the reviewer's concerns have been addressed!

---

### Official Review · Reviewer_kYXT · 2025-10-31

**Soundness:** 3
**Presentation:** 2
**Contribution:** 3
**Rating:** 6
**Confidence:** 4

**Summary:**

This paper introduces EgoDex, a new large-scale egocentric video dataset focused on human dexterous manipulation. Comprising over 300,000 episodes, 90 million frames, and 800 hours of video across 200 tasks and 500 objects, the dataset is meticulously labeled with 3D hand skeleton annotations and calibrated camera parameters. The data collection focused on spontaneous, natural human actions rather than scripted or unnatural movements. This resource is intended to foster advancements in video understanding, embodied AI, and robot manipulation by providing a rich, high-quality foundation for learning dexterity.

**Strengths:**

1. **Largest Scale and Rich Annotation**: EgoDex represents the largest-scale egocentric video dataset of human hand manipulation to date. The inclusion of precise, quantified camera parameters and 3D hand pose labels is a significant contribution, providing an invaluable resource for both video understanding and the development of embodied agents.

2. **Uniquely Naturalistic Data**: The data is collected from spontaneous, active human execution rather than unnatural, deliberately posed or slow-moving scripted actions. This naturalistic quality gives the dataset a unique advantage and is crucial for training robust models that generalize to real-world human behavior.

3. **Novel and Engaging Task Formulation**: The proposed task of predicting human hand trajectories from an egocentric viewpoint is highly intriguing. This task is both academically interesting and potentially beneficial for mutual promotion and integration with robotic motion prediction and control tasks in the future.

**Weaknesses:**

1. **Limited Auxiliary Sensor Data**: The current collection is primarily focused on RGB video. The study could have been significantly enhanced by incorporating additional hardware information during data collection, such as depth maps or synchronized foreground/background segmentation masks. This supplementary data would facilitate lifting observations to 3D and enable subsequent researchers to confidently develop tasks involving procedural background randomization/generation.

2. **Insufficient Detail on Data Collection Protocols**: The manuscript lacks rigor in specifying certain critical aspects of the data collection procedure. Key missing details include the exact number of subjects involved, the strictness of the policy regarding the hand always remaining within the field of view, and a statistical analysis of the duration distribution of tasks. Furthermore, the handling of noisy or erroneous keypoint predictions (e.g., jitter or catastrophic failure) is not discussed.

3. **Inadequate Downstream Task Validation**: The true value of a dataset lies in its utility for downstream tasks. Despite mentioning the positive impact EgoDex could have on robotic manipulation, affordance learning, or video generation, the research does not provide sufficiently solid experimental evidence to prove these claims. For a top-tier conference like ICLR, this lack of rigorous downstream validation significantly weakens the overall persuasiveness of the work.

**Questions:**

1. **Collection Rigor**: Provide exact details on the number of subjects and the protocol for handling and filtering noisy or failed 3D hand pose annotations in the dataset.

2. **Depth/3D Utility**: Conduct a small-scale pilot study demonstrating the utility of the dataset for a 3D-related task (e.g., 3D lifting or affordance modeling) to justify the investment in 3D labels.

3. **Robotics Validation**: Present a clear, quantitative baseline experiment on a robotic manipulation task (e.g., using a SOTA method fine-tuned on EgoDex) to solidify the dataset's direct value to the robotics community.

---

> ### Author Response · Authors · 2025-11-21
>
> We deeply appreciate the time and effort the reviewer has invested in providing us with their constructive comments! We are glad that the reviewer finds the scale and task diversity unprecedented, containing unique data and rich annotation, and with promising applications in many domains. Here is our response to address the remaining concerns:
>
> 1. Limited Auxiliary Sensor Data:
>
> We provide not only RGB video but also paired 3D hand and upper body tracking, camera intrinsics and extrinsics, natural language annotations, and confidence values. This is already more extensive than existing datasets of comparable scale, and the provided modalities are sufficient for training modern vision-language-action (VLA) models. We agree that depth would be a valuable additional modality in the dataset. Unfortunately, power constraints on the Apple Vision Pro do not allow the collection of metric depth with a sufficiently high frequency.
>
> 2. Insufficient Detail on Data Collection Protocols:
>
> Thank you for pointing this out. The data was collected by around 10 operators. Since the operators are instructed to face the table and the Vision Pro has a very wide FOV, the hand(s) performing active manipulation are always within the field of view. For one-handed tasks, the second hand may not always be in the field of view, in which case it is still estimated by the model to be in a reasonable location with respect to the body and given a zero confidence score. Regarding noisy or erroneous keypoints (e.g., due to heavy occlusion), these are also addressed with the provided confidence metrics, which provide detailed per-joint scalar values between 0 and 1. At training time, for instance, a loss mask can exclude the data points with below-threshold confidence values. We will add these details to the final paper.
>
> 3. Inadequate Downstream Task Validation:
>
> We agree that the human motion prediction benchmark does not necessarily translate to downstream policy success rate. The scope of our paper is enabling others in the community to work on the human-to-robot transfer problem without being bottlenecked by the lack of scale and structure in existing human datasets. The downstream integration of this data into control policies is a significant undertaking in its own right, as suggested by recent works focusing specifically on this challenge. There are now several works that focus on improving success rate and generalization by pre-training or co-training robot control policies with the EgoDex dataset introduced in this paper [1,2,3,4]. For example, H-RDT [2] has extensive experimental results that show that pre-training on EgoDex alone increases real-world robot policy success rate by 40.5% over state-of-the-art policies like Pi0 and RDT.
>
> [1] R. Qiu et al. Humanoid Policy ~ Human Policy. Conference on Robot Learning (CoRL), 2025.
>
> [2] H. Bi et al. H-RDT: Human Manipulation Enhanced Bimanual Robotic Manipulation. arXiv preprint arXiv:2507.23523, 2025.
>
> [3] H. Luo et al. "Being-h0: vision-language-action pretraining from large-scale human videos." arXiv preprint arXiv:2507.15597 (2025).
>
> [4] Y. Jiang et al. "RynnVLA-001: Using Human Demonstrations to Improve Robot Manipulation." arXiv preprint arXiv:2509.15212 (2025).

---

### Official Review · Reviewer_cxJH · 2025-11-01

**Soundness:** 3
**Presentation:** 3
**Contribution:** 3
**Rating:** 8
**Confidence:** 4

**Summary:**

This paper introduces a large-scale egocentric manipulation video dataset. It includes comprehensive annotations and is more suitable for robotics task. The data is used to train and evaluate imitation learning policies.

**Strengths:**

1. The paper is well-written and well-organized.
2. The proposed dataset is very useful.
3. The dataset statistics are extensively provided.

**Weaknesses:**

1. Some more pilot experiments should be provided, as mentioned in Section 6. It would be great to see that the downstream experiments are *actually* deployed, not just discussed, to prove the significance of the dataset.
2. More baseline or backbone networks should also be evaluated in the benchmark experiment.
3. Two recent works [1, 2] should be discussed and compared, especially [1].

*Refs*:
[1] MEgoHand: Multimodal Egocentric Hand-Object Interaction Motion Generation. Zhou et al.
[2] TASTE-Rob: Advancing Video Generation of Task-Oriented Hand-Object Interaction for Generalizable Robotic Manipulation. Zhao et al.  CVPR 2025.

**Questions:**

The reviewer believes that some more pilot studies will strengthen the paper's significance, but may not influence its rating to be accepted.

---

> ### Author Response · Authors · 2025-11-21
>
> We thank the reviewer for the strong support and for recognizing the utility and organization of the paper.
>
> 1. Pilot Experiments:
>
> We appreciate the suggestion. While full robot deployment is outside the current scope, the trajectory prediction experiments (Section 5) serve as our primary pilot to validate that the data contains learnable signals. Though we acknowledge that human motion prediction does not necessarily translate to downstream policy success, the primary contribution of this paper is to resolve the data bottleneck in human-to-robot transfer, rather than to solve the control policy itself. The utility of our dataset (EgoDex) for control has been validated by recent works [1-4]; specifically, H-RDT [2] demonstrated a 40.5% improvement in real-world success rates over SOTA models when pre-trained on EgoDex. We maintain that our proposed benchmarks (Trajectory Prediction and Inverse Dynamics) provide a rigorous measure of data quality by isolating it from hardware-specific variables (e.g., kinematics, controller tuning). We appreciate the feedback and will clarify this distinction in the final version.
>
> [1] R. Qiu et al. Humanoid Policy ~ Human Policy. Conference on Robot Learning (CoRL), 2025.
>
> [2] H. Bi et al. H-RDT: Human Manipulation Enhanced Bimanual Robotic Manipulation. arXiv preprint arXiv:2507.23523, 2025.
>
> [3] H. Luo et al. "Being-h0: vision-language-action pretraining from large-scale human videos." arXiv preprint arXiv:2507.15597 (2025).
>
> [4] Y. Jiang et al. "RynnVLA-001: Using Human Demonstrations to Improve Robot Manipulation." arXiv preprint arXiv:2509.15212 (2025).
>
> 2. Missing Citations:
>
> We thank the reviewer for pointing out MEgoHand and TASTE-Rob! We will include these in the Related Work section and add the datasets to Table 1.
>
> We want to highlight that while MEgoHand and TASTE-Rob focuses on interaction and video generation, EgoDex distinguishes itself via the scale of **real** data and the inclusion of precise 3D SLAM-based annotations captured rather than post-hoc reconstruction. EgoDex can be very useful, for instance, as training/evaluation data for the aforementioned works.
>
> 3. Baselines:
>
> We currently evaluate 14 variations of models (including Encoder-Decoder vs. Decoder-only, BC vs. Diffusion vs. Flow Matching). While more variants exist, we believe this set covers the current SOTA architectures for imitation learning policies. In this work, we focused on the most vanilla and simplistic implementations of the representative variations instead of going deeper into optimizing individual VLA models.

---

### Official Review · Reviewer_9Nm4 · 2025-11-02

**Soundness:** 3
**Presentation:** 3
**Contribution:** 3
**Rating:** 6
**Confidence:** 5

**Summary:**

The paper introduces EgoDex, a massive egocentric dataset for dexterous manipulation: ~829 hours, 90M frames, 338k demonstrations, covering ~194 tabletop tasks, recorded at 30 FPS/1080p with paired 3D head/upper-body/hand (25 joints/hand) poses via Apple Vision Pro (with on-device SLAM + calibrated cameras to improve the estimation accuracy). The authors also define two benchmarks—trajectory prediction and inverse dynamics—with a best-of-K evaluation, and report baselines across encoder–decoder vs. decoder-only and BC/DDPM/Flow-Matching models, plus ablations on horizon, goal conditioning, model/dataset size. Visual goal conditioning notably improves final error (−53%), and performance scales with more data. A fixed held-out test split is provided for reproducibility. Lastly, the authors promise to open source all data in the future.

**Strengths:**

The strengths lie in the benchmark design and the extensive effort to collect a well-calibrated dataset. Specifically, they are

- Big and diverse. Way larger than prior human/robot sets, with language, camera extrinsics, and dense dexterity labels—covering ~200 tasks and ~500 objects.

- Clean paired signals. Synced ego RGB + full 3D skeleton (wrists/fingertips/head/arms) at 30 Hz, which is much cleaner than post-hoc hand pose from internet videos.

- Benchmark-ready. Two well-defined tasks (trajectory prediction, inverse dynamics) with a fixed held-out test split, so results are easy to compare apples-to-apples.

- Useful takeaways. The experiments surface a few clear patterns—for example, goal conditioning helps consistently.

**Weaknesses:**

The weaknesses include:

- Benchmark scope is narrow. The evaluation focuses on human motion prediction only, without assessing the robot side (e.g., retargeting quality and policy performance). Human trajectory prediction is only an intermediate signal; good imitation error does not necessarily translate to task success—this disconnect has been noted before [1], and that’s based on robot data—all the more so for human trajectory prediction as the retargeting might also amplify the error. A more direct metric like robot task success rate (in sim or on hardware) would better capture end-to-end utility. I agree trajectory metrics are reproducible, but they’re not decisive for manipulation.

[1]. Mandlekar, Ajay, et al. "What matters in learning from offline human demonstrations for robot manipulation." arXiv preprint arXiv:2108.03298 (2021).

- Practicality of goal images. Minor but important: how is the goal image obtained at inference? If it’s required, please discuss practical acquisition (e.g., user-provided snapshot, retrieval from a library, last-frame of a setup video) and report results (or at least ideas and any preliminary explorations) with vs. without a goal image to gauge real-world usability during inference.

**Questions:**

See Weaknesses.

---

> ### Author Response · Authors · 2025-11-21
>
> Thanks to the reviewer for the constructive and encouraging feedback! We appreciate that the reviewer acknowledges the scale, diversity, detailed annotations, benchmarks, and insights we make available through this work. To address the remaining concerns, here is our response:
>
> 1. Benchmark scope:
>
> We fully agree that the human motion prediction benchmark does not necessarily translate to downstream policy success rate. The scope of our paper is enabling others in the community to work on the human-to-robot transfer problem without being bottlenecked by the lack of scale and structure in existing human datasets. The downstream integration of this data into control policies is a significant challenge in itself, as suggested by recent works focusing specifically on this challenge. There are now several works that focus on improving success rate and generalization by pre-training or co-training robot control policies with the EgoDex dataset introduced in this paper [1,2,3,4]. For example, H-RDT [2] has extensive experimental results that show that pre-training on EgoDex alone increases real-world robot policy success rate by 40.5% over state-of-the-art policies like Pi0 and RDT.
>
> Despite the limitations of human motion prediction benchmark, we believe the proposed benchmarks (Trajectory Prediction and Inverse Dynamics with best-of-k evaluation) are a rigorous way to measure the quality of the learned representations without introducing the confounding variables of specific robot kinematics, hardware, controller tuning, and retargeting noise. Still, we appreciate the reviewer's constructive feedback and will incorporate this discussion in the final version.
>
> [1] R. Qiu et al. Humanoid Policy ~ Human Policy. Conference on Robot Learning (CoRL), 2025.
>
> [2] H. Bi et al. H-RDT: Human Manipulation Enhanced Bimanual Robotic Manipulation. arXiv preprint arXiv:2507.23523, 2025.
>
> [3] H. Luo et al. "Being-h0: vision-language-action pretraining from large-scale human videos." arXiv preprint arXiv:2507.15597 (2025).
>
> [4] Y. Jiang et al. "RynnVLA-001: Using Human Demonstrations to Improve Robot Manipulation." arXiv preprint arXiv:2509.15212 (2025).
>
> 2. Practicality of Goal Images:
>
> We agree that it may not be always practical to obtain a goal image at deployment time. Including a goal image is not necessary for training and deploying policies with EgoDex data. The goal image is only included in a visual goal-conditioning experiment (1 of 14 models trained) to show how richer conditioning signals help reduce the multimodality of natural human motion.
>
> As for how goal images are obtained, in our Inverse Dynamics benchmark, the goal image is available as the last frame of the action chunk in held-out test trajectories. In practical deployments, some ways this may be achieved include:
>
> * Retrieval: Retrieving a goal frame from a library of successful demonstrations [5].
>
> * Image generation or inpainting with user prompt: given the instruction, use an image inpainting model to generate the goal frame [6].
>
> We will add a discussion on these inference-time modalities to the final paper. We also note that our Table 4 results show that even without goal images, the models perform reasonably well, though goal conditioning significantly tightens the trajectory.
>
> [5] Myers, Vivek, et al. "Goal representations for instruction following: A semi-supervised language interface to control." Conference on Robot Learning. PMLR, 2023.
>
> [6] Nair, Ashvin V., et al. "Visual reinforcement learning with imagined goals." Advances in neural information processing systems 31 (2018).

---

### Meta-Review · Area_Chair_iDuC · 2026-01-07

**Summary:**

The paper got generally positive ratings, 6,8,6,and 4.

Reviewers commonly recognized the importance, novelty, and advantages of the proposed dataset.

As major concerns, reviewers noted that providing more detailed descriptions of the dataset and additional validations of its advantages on downstream tasks (such as robotics applications) would further strengthen the paper’s impact.

**Reviewer Concerns:**

In general, the authors' response provides convincing answers to reviewers' concerns.

**Reviewer Scores:**

During the reviewer-author discussion phase, the reviewer who initially gave a negative rating indicated an intention to increase their final score.

The AC expects that the other reviewers’ ratings will remain unchanged, as they already provided positive evaluations.

The AC largely agrees with the reviewers’ comments. The proposed dataset should be helpful and impactful for the community, placing the paper above the acceptance bar.

However, the paper could be further strengthened through more rigorous analysis of the proposed dataset and additional experiments on the presented prediction tasks, or by including benchmarks that demonstrate other downstream applications. At present, the experimental section does not appear sufficiently strong, and there remains room for improvement to further enhance the overall quality of the paper.

---

### Decision · Program_Chairs · 2026-01-26

Accept (Poster)